# Musculoskeletal Pain in the Neck and Lower Back Regions among PHC Workers: Association between Workload, Mental Disorders, and Strategies to Manage Pain

**DOI:** 10.3390/healthcare11030365

**Published:** 2023-01-28

**Authors:** Marta Regina Cezar-Vaz, Daiani Modernel Xavier, Clarice Alves Bonow, Jordana Cezar Vaz, Letícia Silveira Cardoso, Cynthia Fontella Sant’Anna, Valdecir Zavarese da Costa, Carlos Henrique Cardona Nery, Aline Soares Alves, Joice Simionato Vettorello, Jociel Lima de Souza, Helena Maria Almeida Macedo Loureiro

**Affiliations:** 1School of Nursing, Federal University of Rio Grande, Rio Grande 96203-900, Brazil; 2Faculty of Nursing, Federal University of Pelotas, Pelotas 96010-610, Brazil; 3Institute of Dermatology Professor Rubem David Azulalay (Medical Residency), Rio de Janeiro 20020-020, Brazil; 4Department of Nursing, Federal University of Pampa, Uruguaiana 97501-970, Brazil; 5Department of Nursing, Federal University of Santa Maria, Santa Maria 97105-900, Brazil; 6Institute of Human and Information Sciences—ICHI, Federal University of Rio Grande—Santa Vitória do Palmar Campus, Santa Vitória do Palmar 96230-000, Brazil; 7School of Nursing (Ph.D. Program), Federal University of Rio Grande, Rio Grande 96203-900, Brazil; 8School of Health Sciences, Santiago University Campus, University of Aveiro (ESSUA), Aveiro 3810-193, Portugal

**Keywords:** health personnel, musculoskeletal pain, lower back pain, neck pain, working conditions, workload, mental disorders, primary health care

## Abstract

Scientific evidence indicates that workers in the health sector are commonly exposed to work-related musculoskeletal pain. Objectives: We aimed to identify the relationship between the presence and intensity of musculoskeletal pain in the neck and lumbar regions reported by Primary Health Care (PHC) workers with workloads and occupational risks, analyze musculoskeletal pain in the presence and absence of self-reported mental disorders based on a medical diagnosis, and identify workers’ strategies to manage pain. Method: This cross-sectional study addressed 338 health professionals working in PHC outpatient services in the extreme South of Brazil. One questionnaire addressed sociodemographic questions concerning occupation, occupational risks, and mental disorders. The Nordic Musculoskeletal Questionnaire was used to assess self-reported musculoskeletal pain. The National Aeronautics and Space Administration Task Load Index (NASA-TLX) measured the workload. A descriptive and inferential analysis was performed using SPSS version 21.0. Results: Most (55.3%) participants reported neck and (64.5%) lower back pain in the previous 12 months, and 22.5% and 30.5% reported intense neck and lower back pain, respectively, in the previous 12 months. The results showed different independent associations with increased musculoskeletal pain among health workers. Dentists presented the highest prevalence of neck pain, while female workers presented the highest prevalence of lower back pain. Furthermore, the perception of ergonomic risk and virtually all self-reported mental disorders (except panic syndrome for neck pain) were associated with pain in the neck and lower back regions and a higher frustration level (mental demand). Additionally, professionals with graduate degrees, nurses, and professionals working the longest in PHC services reported seeking complementary therapies more frequently, while physicians and those with self-reported mental disorders self-medicated more frequently.

## 1. Introduction

Musculoskeletal pain in the back region is one of the leading causes of occupational absenteeism, which can entail considerable costs for the public health system [1] in terms of the possible consequences on the biopsychosocial well-being of affected workers as well as human resource deficits that can affect the quality of work processes. In a narrative review of the literature on the global burden of conditions related to musculoskeletal pain, the authors [2] reinforced that the Global Burden of Disease Study confirmed that musculoskeletal pain contributed to the global burden of disability of youth and adults. They also commented that such evidence did not support effective global policy initiatives [2]. 

In this complex context of multifaceted conditions and different biopsychosocial conditions, it should be noted that musculoskeletal pain can represent different musculoskeletal conditions [2]. However, the present study focused on the musculoskeletal condition of back pain, specifically in the neck and lower back regions. Concerning this particular focus, “recent GBD 2016 estimates showed that low back pain was the leading cause of years lived with disability in most countries and territories, and musculoskeletal conditions as a group were the leading burden of noncommunicable disease-related disability” [2].

In this global scenario, from the study on the Global Burden of Disease Study carried out by [3] on the global burden of disease, it was shown that lower back pain is the principal cause of years lived with disability and the third leading cause of disability-adjusted life years in Brazil. Thus, neck pain represented the sixteenth clinical condition with the most significant impact on the population’s health in 2019 [4]. According to these authors [4], the Brazilian population is in an “accelerated aging process,” and prevention strategies for these diseases must be developed throughout life, with changes in personal and behavioral habits, as well as the expansion of adequate access to health care for these specificities [4]. 

Furthermore, in this broader context, people spend much of their lives in work environments, which are determinants of health and disease states. Moreover, based on evidence, musculoskeletal conditions are known as producers of disability burden and are directly related to the aging process of people worldwide. Furthermore, this evidence includes, in addition to personal/behavioral risk factors, those related to work [2]. Previous studies on musculoskeletal pain in workers of different jobs showed an association of different risk factors for developing these painful conditions, mainly in the back, but not restricted to this area [5,6]. 

Musculoskeletal pain in the neck and lower back commonly affects health workers and is directly linked to occupational conditions such as physical and psychological workloads [7,8]. Personal characteristics also influence the incidence of musculoskeletal disorders, including lower back pain and neck pain; for instance, gender, marital status, education, body mass index (BMI), income, activities performed outside of work, and exercise (including non-work-related exercise) [9]. Psychosocial factors, such as anxiety and stress [10,11], and workload endured during a workday also play a role. Furthermore, women are considered to be more frequently affected by musculoskeletal pain than men [9,12].

Therefore, the scientific world has proven that musculoskeletal pain among health workers represents a severe global public health problem that leads to work-related disabilities. Disabilities can be caused by accidents at work or even cause them, resulting in considerable financial consequences, such as those related to expenses with therapeutic services, and impairing the quality of life of the individual and their family. In addition, several work-related factors predispose individuals to these disorders, such as physical and psychosocial workloads and personal and occupational characteristics, such as years of experience in the health sector [9,13,14]. 

Most studies addressing the occupational health of health workers focus on hospital settings [10,15,16,17,18,19,20], considering work process characteristics. Few studies address musculoskeletal health among Primary Health Care (PHC) workers [21,22,23,24,25,26], which is the object of this study. Evidence provided in the literature indicates that such disorders condition the work and lives of those affected. Studies show that healthcare workers are usually exposed to musculoskeletal pain. Thus, protecting the health and safety of these workers contributes to improving their productivity, job satisfaction, and employee retention. Poor working conditions leading to work-related diseases, injuries, and absenteeism impose high costs on the health sector [27]. 

Given this, work organizations must promote healthy conditions for the workforce, including conditioning elements that group the most diverse biopsychosocial factors possible to promote healthy and safe environments for people that support work processes. However, for the best decisions to be made at the political and operational level, it is necessary to constantly produce evidence generated in the workplace with a focus on workers and working conditions to strengthen global evidence. The few studies found in the literature on musculoskeletal pain in PHC workers did not integrate the conceptual elements proposed in the present study, that is, the key areas of focus concerning musculoskeletal pain in PHC workers in our study include the workload, occupational risks, and mental disorders, in addition to work and personal characteristics. This set of elements comprises the most comprehensive characteristics of musculoskeletal pain in the back region when considering the relationship with the work environment of the PHC services. Hence, this study’s objective was to identify the relationship between the presence and intensity of musculoskeletal pain and lumbago regions among PHC workers and exposure to workloads and occupational risks. Additionally, musculoskeletal pain was analyzed in the presence and absence of mental disorders (based on a medical diagnosis). Finally, workers’ strategies to manage neck and lower back pain were also identified.

## 2. Materials and Methods

This study integrated the macro project “Dimensão Socioambiental da Saúde dos Trabalhadores da APS do Sul do Brasil” (The Socio-Environmental Dimension of the Health of PHC Workers in the South of Brazil). The National Council for Scientific and Technological Development (CNPq) provided financial support. This study was part of the macro project’s first phase, which has already been completed, while the other stages are still in progress. A research team is responsible for this project, which is being conducted in the Laboratory of Socio-environmental Process Studies and Collective Health Promotion (LAMSA), which itself is linked to several universities in Southern Brazil and abroad. It should be noted that the methodological elements and procedures in this section were described in our previous studies with the same population and other outcomes [28,29].

### 2.1. Study Design and Participants

This cross-sectional study was conducted in two cities in the extreme South of Rio Grande do Sul, Brazil. City 1 has medium-sized characteristics and 31 outpatient PHC units, while City 2 is small and has 10 PHC outpatient units. The sample size was calculated using the Epi Info^®^ StatCalc tool (version 7.2, CDC, Atlanta, GA, USA). During the study period, 548 typical PHC workers following the National Health Policy of the Brazilian government [30] were considered, including nurses, physicians, dentists, nursing technicians/assistants, community health agents, and oral health technicians/assistants. A margin of error of 5%, a 95% confidence level, and losses of 5% were established. The professionals were selected using non-probabilistic sampling. The consecutive intentional sample [31] was supposed to comprise at least 232 professionals from the covered area. The inclusion criterion was working in a PHC service for at least six months, and the exclusion criterion was being on leave during data collection (from January to March 2020). A total of 342 health professionals were interviewed [28]. However, specifically in this study, 338 interviews were included in the analysis because 4 interviews were incomplete. The participants were distributed as follows: 50 nurses, 43 physicians, 72 nursing technicians, 139 community health agents, 13 dentists, 15 oral health technicians/assistants, and 6 others. The others were professionals within the PHC health teams that complement the typical PHC teams according to the local organization following the National Health Policy of the Brazilian government [30], and they were included in the study to consider the characteristics of these health teams. This decision was based on work similarities and differences; both aspects were carefully addressed in data analysis.

PHC workers were recruited and interviewed for this project stage from January to March 2020. Face-to-face interviews were held at their workplaces by previously trained researchers. Two or three researchers always worked together to ensure their safety and speed up the selection process. At the time, teaching and research activities had not yet been suspended due to the COVID-19 pandemic. For this reason, the individual interviews with the PHC workers were face-to-face and lasted 58 min on average. All participants received clarification about the study’s objectives and signed two copies of informed consent forms. Additionally, STROBE guidelines were complied with [32].

### 2.2. Measures

Four questionnaires were used in this study to collect data. First, a structured form addressed sociodemographic information (i.e., age, self-reported race, marital status, education, the number of children, and body mass index BMI = weight/height ^2^) and information regarding PHC work (i.e., place of work, whether the participant had a second job, profession, years of professional experience, years working in a PHC service, weekly working hours, work shift at the PHC facility, and monthly income). This same structured questionnaire included questions addressing exposure to occupational risks at the PHC workplace [33] and the presence/absence of mental disorders [34,35] (i.e., anxiety disorder, depressive episode, acute stress reaction, nonorganic disorder of the sleep–wake schedule, and panic disorder). Next, the Nordic Musculoskeletal Questionnaire (NMQ) [36] was adopted to investigate pain in the regions selected for this study, i.e., neck and lower back pain, followed by a 10-point visual analog scale (VAS) to assess pain intensity [37]. Then, the National Aeronautics and Space Administration (NASA) Task Load Index (TLX) (NASA-TLX) [38,39] was used to assess the subjective workload in PHC services. Finally, the workload was considered to aggravate pain among PHC workers.

The Nordic Musculoskeletal Questionnaire (NMQ) is a validated tool used to investigate musculoskeletal symptoms in nine body sites [36]. The questionnaire evaluates the presence of pain/discomfort within the previous 12 months and the previous 7 days, functional impairment, and whether the individual has sought health care within the previous 12 months; the answers are dichotomous. The questionnaire was validated in Portuguese [40,41] and is widely used in occupational health, contributing to identifying workers with pain in different regions of the body. In addition, the NMQ presented satisfactory reliability (Cronbach’s α = 0.917).

Mental disorders were assessed considering the PHC professionals’ self-reported medical diagnoses that met the definitions of the Ministry of Health [42] and the ICD-10, according to the World Health Organization (WHO) [34]. As a result, the following were found: (1) Generalized anxiety disorder (F41.1)—excessive and persistent anxiety, though it is not restricted to, or strongly predominates in, any particular circumstance (i.e., “free-floating”). Dominant symptoms vary but may include persistent nervousness, trembling, muscle tension, sweating, lightheadedness, palpitations, dizziness, and epigastric discomfort; (2) Depressive episode (F32)—patients experience a low mood and exhibit reduced energy, decreased activity, loss of interest, and reduced concentration. Sleep is usually disturbed, and appetite is diminished. There are also feelings of guilt or worthlessness; (3) Acute stress reaction (F43.0)—this disorder concerns a transient disorder that develops without any other apparent mental disorders in response to exceptional physical and mental stress and usually subsides within hours or days. Its symptoms include an initial state of a “daze” with some constriction of the field of consciousness, narrowing of attention, inability to comprehend stimuli, and disorientation; (4) Nonorganic disorder of the sleep–wake schedule (F51.2)—characterized by a lack of synchrony or a poor sleep–wake cycle, resulting in either insomnia or hypersomnia; (5) Panic disorder (episodic paroxysmal anxiety disorder) (F41.0)—characterized by recurrent, severe, and unpredictable anxiety (panic) attacks. Dominant symptoms include palpitations, chest pains, choking sensation, dizziness, and feelings of unreality.

The universal concept of the Occupational Health and Safety Act was used to analyze the workers’ perception of exposure to occupational risks [33], which classifies occupational risks as physical, chemical, biological, physiological, or psychosocial [43]. It is noteworthy that these concepts were integrated with other concepts in our previous study [29].

Finally, the National Aeronautics and Space Administration Task Load Index (NASA-TLX) [38] was designed to capture the workers’ subjective experience [38,39] and has been applied in different work environments [44,45,46]. Additionally, this research group has already used on this scale in previous studies [47,48] and investigated other phenomena in the same population addressed in this study [28]. The workload in the PHC services was verified according to the following demands: mental (mental and perceptive activities required by tasks, such as thinking, calculating, remembering, looking, researching, and decision-making, among others), physical (physical efforts, such as walking, pushing, pulling, turning, sliding, and controlling, among others), temporal (time required to perform a given task, and whether work pace is slow or fast), performance (quality and agility with which tasks are performed, i.e., how successful workers believe they are when performing tasks), total effort (mental and physical effort needed for workers to keep their performance levels), and frustration level (feeling of insecurity, discouragement, and irritation caused by work tasks) [39]. Based on these 6 domains, the health workers were asked to rate workloads from 0 to 20 (0 = no demand and 20 = most intense demand) [39]. According to the NASA-TLX guidelines, the results concerning the workload assessment are classified into four demand levels: scores from 0–5 refer to a low level; 6–10, moderate–low level; 11–15, moderate–high level; scores from 16–20 refer to a high-level demand. Further details concerning the professionals’ workload are provided in our previous study [28]. In this study, the reliability of NASA-TLX verified through Cronbach’s alpha was considered acceptable (α = 0.767).

Additionally, multiple-choice questions were included to address the PHC workers’ strategies used to manage neck and lower back pain, such as (1) complementary therapies, where the respondents could select aromatherapy, acupuncture, chiropractic, Reiki, meditation, and/or auriculotherapy, among others; (2) self-care, including exercises such as gymnastics, walking outdoors, stretching, body relaxation, and muscle strengthening; (3) self-medication, concerning the taking of allopathic medications, mostly analgesics and anti-inflammatories; (4) seeking an Emergency Room, which could be selected in the presence of intense pain or other musculoskeletal disorders; (5) seeking a specialist, an option for those who sought a physician and/or physical therapist or other professional trained to provide care for musculoskeletal disorders; (6) seeking a healthcare unit, an option for those who sought a PHC unit. In this case, most health teams include members with family health training and provide integral care for different diseases and conditions affecting the population in the covered communities.

The structured questionnaire’s variables (i.e., sociodemographic and occupational variables, PHC workload, exposure to occupational risks, and dichotomous variables concerning mental disorders, i.e., anxiety disorder, depressive episode, acute stress reaction, nonorganic disorder of the sleep–wake schedule, and panic disorder) and the PHC workers’ strategies to manage neck and lower back pain were tested and adjusted during meetings held at the Laboratory for the Study of Socioenvironmental Processes and Collective Health Production (LAMSA). A pilot study was also performed with a sample of ten individuals from different professions before data collection, which supported this process. The objective was to assess and adapt the data-collection instrument regarding the effectiveness of its application and verify the participants’ cognitive understanding and whether they considered the questions easy or difficult, as well as training the field researchers [28,29]. Figure 1 outlines the key concepts and operational qualifiers adopted in this study.

### 2.3. Data Analysis

The quantitative variables were described according to the mean and standard deviation (symmetric distribution) or median and interquartile range (asymmetric distribution), depending on the variables’ distribution. The Kolmogorov–Smirnov normality test determined the type of distribution. Categorical variables were described using absolute and relative frequencies. The Student’s t-test for independent samples was used to compare the means between workers with and without the outcome (neck or lower back pain). The Mann–Whitney test was used for asymmetric cases (outliers), and Pearson’s Chi-square or Fisher’s Exact test was performed to assess associations between the categorical variables. Adjusted residual analysis was used to locate the differences in polytomous variables when the Chi-square test presented statistical significance.

Poisson Regression was used to control for confounding factors. The criterion for the variables to enter the model was presenting a *p*-value of <0.20 in the bivariate analysis, while a *p*-value of <0.10 was needed to remain in the final model. The prevalence ratio was calculated with a 95% confidence interval (Appendix A). The significance level was set at 5% (*p* ≤ 0.05), and the analyses were performed using SPSS version 21.0 (IBM Corp., Armonk, NY, USA).

All participants received clarification regarding the study’s objectives and signed two copies of the consent form before entering the study, which followed the Declaration of Helsinki (reviewed in 2013) (WMA—The World Medical). The protocol was approved by the Research Ethics Committee (Conep) at the Federal University of Rio Grande (CAAE: 70043717.0.0000.5324).

## 3. Results

### 3.1. Descriptive and Bivariate Analysis

A total of 338 PHC workers were included in this study. They included integrated multidisciplinary teams of Primary Health Care (PHC) outpatient services located in cities in the extreme South of Brazil. City 1 is medium-sized and hosts 31 PHC outpatient units, while City 2 is small and hosts 10 PHC outpatient units. The participants were 41.4 (±9.9) years old on average, most were women (86.6%), married or in a consensual union (58.6%), self-reported Caucasian (77.2%), and (81.6%) did not have a paid job besides the one in the PHC service. Additional characteristics of PHC workers can be observed in our previous study [29]. The analysis revealed that 55.3% of the PHC workers reported neck pain within the previous 12 months, which impeded 21.9% from performing their tasks due to pain. Furthermore, 26% of PHC workers reported pain or discomfort in the neck within the previous 7 days. The median pain intensity found with the 10-point VAS was 3 points (25–75 percentiles: 0–7), while 22.5% of PHC workers reported intense neck pain within the previous 12 months (≥8 points). In addition, a higher prevalence of lower back pain was found; 64.5% of workers reported it within the previous 12 months, which impeded 27.8% from performing work tasks. Within the previous 7 days, 32.2% of PHC workers experienced lower back pain and the median of pain intensity, according to the VAS, was 5 points (25–75 percentiles: 0–8). A total of 30.5% of PHC workers reported intense lower back pain within the previous 12 months (≥8 points). Hence, musculoskeletal pain increased within the previous 7 days. Additionally, a trend was found for the number of professionals with intense neck pain being unable to work within the previous 12 months. On the other hand, the relationship between the number of workers with intense lower back pain and an impediment to performing job tasks was not very clear.

Additionally, 80% and 75.9% of PHC workers with neck and lower back pain, respectively, reported that their conditions were directly related to their work at the PHC service. They acknowledged that their pain was a vital sign of musculoskeletal disorder caused by physical and psychological conditions of the PHC work environment.

Table 1 presents the associations between variables and neck and lower back pain within the previous 12 months. This set of variables includes sociodemographic (i.e., age, gender, self-reported race, marital status, education, number of children, professionals’ BMI, and monthly income) and occupational variables (i.e., the city where the service is located, having a second job besides the PHC service, type of unit, profession, years of professional experience, years working at the PHC center, weekly workload, working hours at the PHC center, and occupational risks). There are also the subsets concerning workload dimensions (mental demands, physical demands, temporal demands, performance, total effort level, and frustration level) and the presence or absence of mental disorders (e.g., anxiety, depressive episode, panic syndrome, stress, and sleep–wake cycle disorders).

Significant associations were found with the following variables and pain in both the cervical and lumbar regions: more experience in PHC services (*p* = 0.007; *p* = 0.047); higher workload scores concerning mental demands (*p* = 0.015; *p* = 0.024) and frustration level (*p* < 0.001; *p* = 0.014); greater perception of physical (*p* = 0.012; *p* = 0.001), chemical (*p* = 0.028; *p* = 0.002), and ergonomic occupational risks (*p* = 0.001; *p* < 0.001); mental disorders (*p* < 0.001; *p* < 0.001). Almost all mental disorders were associated with neck and lower back pain, except panic syndrome and neck pain. Differences between gender were found only for lower back pain in this bivariate analysis; a greater percentage of female PHC workers with lower back pain was found (*p* = 0.012). Additionally, significant differences were found between lower back pain and high BMI (*p* = 0.043) and the perception of biological risk (*p* = 0.010) (Table 1).

### 3.2. Multivariate Poisson Regression Analysis

After the analysis to control for confounding factors, the variables that obtained a *p*-value < 0.20 in the bivariate analysis were included in the multivariate Poisson Regression model. However, only variables with a *p*-value < 0.10 remained in the final model (Table 2). The following independent factors remained statistically associated with neck pain reported within the previous 12 months: dentists (*p* = 0.014), more years of experience in PHC services (*p* = 0.038), higher frustration level (*p* = 0.005), being exposed to ergonomic occupational risks (*p* = 0.026), and mental disorders (*p* < 0.001). These results show that dentists had a prevalence of neck pain within the previous 12 months 67% higher than that of physicians (RP = 1.67; 95%CI: 1.11–2.51). Although statistical significance was recorded for the variable of other PHC workers, caution should be exercised with this result, especially given the number of participants in the sample. Additionally, PHC workers were 2% more likely to experience neck pain within the previous 12 months for every extra year working in the PHC service (RP = 1.02; 95%CI: 1.00–1.03). Another influential factor that remained was the level of frustration, i.e., the prevalence of neck pain increased by 3% for every extra point in frustration level (RP = 1.03; 95%CI: 1.01–1.04). Furthermore, the prevalence of neck pain among PHC workers who perceived ergonomic occupational risks increased by 67% (RP = 1.67; 95%CI: 1.06–2.62). Finally, PHC workers reporting mental disorders were 88% more likely to experience neck pain (RP = 1.88; 95%CI: 1.39–2.54). 

The following variables remained associated with lower back pain reported within the previous 12 months: being a female PHC worker (*p* = 0.042), perceived exposure to ergonomic occupational risks (*p* = 0.025), and mental disorders (*p* = 0.001). BMI and chemical occupational risks were close to the significance threshold after adjustment (*p* = 0.073 and *p* = 0.070, respectively). Therefore, female PHC workers were 40% more likely to report lower back pain within previous 12 months than their male counterparts (RP = 1.40; 95%CI: 1.01–1.94). The prevalence of lower back pain increased by 62% among those perceiving ergonomic occupational risks (RP = 1,62; 95%CI: 1.06–2.48). Finally, PHC workers with mental disorders were 45% more likely to report lower back pain (RP = 1.45; 95%CI: 1.16–1.82). 

### 3.3. Analysis Concerning PHC Workers’ Strategies to Manage Neck and Low Back Pain

Most PHC workers reported self-care (e.g., light walking, stretching, body relaxation), and almost half (>45%) reported self-medication (allopathic and/or homeopathic medications), regardless of the pain site (Figure 2). 

After the multivariate analysis, all the variables with significant associations were tested to identify their potential association with PHC workers’ strategies to manage neck and lower back pain. Female PHC workers sought the PHC service to care for musculoskeletal pain more frequently than men (*p* = 0.040) (Appendix A). In addition, those with a graduate degree more frequently resorted to complementary therapies (e.g., aromatherapy, acupuncture, chiropractic, Reiki) (*p* = 0.032), while PHC workers with up to high school education more frequently sought the PHC service (*p* = 0.003) (Appendix A).

Furthermore, nurses more frequently sought complementary care (*p* = 0.002), physicians self-medicated more (*p* = 0.029), and nursing technicians/assistants, dentists, and oral health technicians/assistants more frequently sought a specialist (*p* = 0.002). In contrast, community health agents and oral health technicians/assistants usually sought a PHC service (*p* < 0.001) (Appendix A).

Professionals dealing with mental disorders self-medicated more frequently (*p* < 0.001) and less frequently sought care procedures (*p* = 0.048) (Appendix A). Workers with more years of experience in PHC services (≥8 years) resorted more to complementary care (*p* = 0.005) and sought a specialist (*p* = 0.041) (Appendix A). 

## 4. Discussion

The results show that most PHC workers reported not only neck and lower back pain within the previous 12 months but also high-intensity pain. However, the prevalence of neck pain was higher among dentists. Studies [49,50] show that neck pain results from excessive static stretching loading with sustained muscle activity of the sternocleidomastoid or trapezius muscle. It is a common work-related musculoskeletal disorder among dentists [49,50]. Therefore, immediate corrective measures are recommended among PHC services. Such measures should include adequate staffing to decrease patient–dentist relationships, encouraging ergonomic postures, fewer working hours, and promoting stress relief. There should also be adequate working conditions to decrease physical risks, such as exposure to machinery noise, vibration, poor or excessive lighting, and humidity, as well as exposure to chemicals such as amalgamators, and disinfectants such as alcohol, glutaraldehyde, sodium hypochlorite, and chlorhexidine, in addition to medical gases (e.g., nitrous oxide) [51]. Furthermore, dentists should be adequately trained and educated regarding ergonomics and the benefits of exercising to decrease frustration arising from workload [51,52,53].

In addition to dentists, nurses also reported a high prevalence of neck pain. Studies [54,55] performed in Poland with nurses show that the origin of neck pain may be related to occupational factors such as a high-paced work environment, repetitive movement patterns, insufficient time to recover, weight lifting, inadequate postures, mechanical pressure, flexion, torsion, vibrations, and low temperatures [54]. Although we did not identify the particularities of postures and programmed and regular physical exercises in the present study, studies mentioned previously include in their recommendations self-protecting techniques such as maintaining good posture, adequate chair support, resting between treatments, regular back-strengthening exercises, and adequate rehabilitation [51,52,53].

Regarding associations between the participants’ gender and pain, female PHC workers presented a higher prevalence of lower back pain within the previous 12 months than their male counterparts. One study conducted in Uganda with health workers from hospital services showed that the main factors of occupational risk for lower back pain include [56] lifting and moving patients, frequent twisting and bending, sustained postures, poor ergonomics in the work environment, anxiety, depression, stress, poor job satisfaction, inadequate staffing, and poor working conditions. A previous study showed a high frequency of lower back pain among female health workers, possibly explained by the extra working load among women, such as domestic chores and anatomic, physiological, and structural differences between women and men. Such differences include feminine hormones, e.g., pregnancy-induced relaxin and low estrogen levels associated with the aging process, which aggravates tension in the spine bone [56]. A universal recommendation is for health workers to seek a balance between caring for their spines and the care provided to patients who require manual handling. Therefore, ergonomic structuring, work organization, medical care training, and auxiliary devices can reduce these occupational risks [56,57,58].

It is important to note that PHC workers with the longest working time in PHC service presented a higher prevalence of neck pain. Previous studies corroborated this finding. For example, studies [23,59] conducted in Brazil show that because PHC units are the main entrance door to the Unified Health System (SUS), they are stressful and tension-laden environments for health teams. Such musculoskeletal “wear and tear” aggravate over time, triggering musculoskeletal morbidities among PCH workers, with neck pain being a highly prevalent clinical outcome. The reason is that the work process is exhausting, with repetitive movements and incorrect postures, causing disabilities among PHC workers and compromising their quality of life and patient care [23,59]. Likewise, other studies show the relationship between work demands in PHC services, years of experience, and musculoskeletal pain, especially in the neck region [22].

This study shows that given the characteristic work performed by PHC workers or the way work is imposed, frustration levels caused by the workload is an aggravating factor for neck pain. Studies carried out in Malaysia [60] with hospital service nurses, and in Portugal [21] with PHC nurses, showed that the lack of investment in the health of workers in these health services, combined with poor working conditions, resulted in work overload and multiple causes of musculoskeletal disorders, such as neck pain, which, as presented here, can aggravate the level of frustration. This condition affects the well-being of health workers due to the discomfort caused by pain in the work environment. As a result, frustration is directly related to mental conditioning caused by work and time pressure. These studies also show that PHC services with inadequate structure and management may psychologically and physically harm health workers, possibly compromising patient care and workers’ problem-solving capacities.

Consequently, this study highlights that ergonomic occupational hazards are potential occupational stressors, triggering cervical and lumbar pain. In addition, studies conducted in India [61] (on dentists), Sweden [62] (studies with health workers and social care workers were reviewed), and Georgia [63] (with surgeons, nurses, and dentists) showed several risk factors for musculoskeletal pain among healthcare workers, including individual characteristics, poor working posture, manual handling of heavy loads, repetitive motions, strenuous exertion, job stress, and long working hours. In these work contexts, a wide range of occupational stressors can negatively affect the health of PHC workers, including psychosocial factors that can intensify musculoskeletal pain, as demonstrated in the present study. 

Although the present study does not cover work ergonomics, it is relevant to prioritize a multifaceted approach in work environments [63], which includes equipment, ergonomics, training workers on how to deal with patients, and providing focused exercise programs in strength training to decrease musculoskeletal pain, as well as the risk of musculoskeletal disorders [64]. These strategies can promote the health status of professionals and healthier work environments. 

Additionally, the workers’ neck and lower back pain were significantly associated with mental disorders. Hence, this study shows that mental overload experienced by PHC workers and musculoskeletal disorders, especially neck and lower back pain, favor mental disorders such as anxiety, depression episode, stress, and sleep–wake cycle disorder. On the other hand, panic syndrome only appears to influence lower back pain. Similar results are found in the literature [65], indicating the need to promote health and prevent work-related disorders, as neck and lower back pain lead to absences from work and exacerbate mental disorders [66].

Finally, the results show that the workers presenting a mental disorder self-medicated to relieve pain. Physicians were the ones who self-medicated most frequently [67], while nurses more frequently sought integrative and complementary care [68]. 

### Limitations and Lines of Research

This study’s limitations are due to its cross-sectional design, as it does not enable the establishment of direct causal relationships between the variables. Musculoskeletal pain in the neck and lower back regions may aggravate and lead to mental disorders. However, this study’s cross-sectional design did not allow for measuring the event’s temporality and effect, i.e., whether anxiety, stress, or sleep–wake cycle disorder, for instance, occurred during or after the workers experienced pain. Therefore, future studies should include longitudinal designs. Additionally, although the sample size was adequate, intentional sampling was adopted, and the study was conducted in only two cities. Therefore, the results should be cautiously interpreted and generalized [29]. Another limitation is that few studies address this population, impeding comparisons between musculoskeletal pain and a historical series. Finally, this study was performed before the COVID-19 pandemic; hence, its results can collaborate with studies conducted during and after the pandemic, enabling comparisons considering the outcomes addressed here. In this sense, this study can support future research and in-depth analyses concerning working conditions and the specificity of musculoskeletal pain affecting healthcare providers they consider to be work-related. Studies should assess the risks of self-medication, pain intensity, and pain relief by maintaining or changing workers’ strategies to manage pain. Such studies can support interventions in which actions are adjusted with those interested as well as carefully considering elements in the workers’ routine and improving individual and collective conditions. Furthermore, this study can support managers in planning and implementing protective, preventive, and care measures for professionals working in PHC outpatient services.

## 5. Conclusions

This study’s results indicate an association between the factors related to profession, length of experience in a PHC service, occupational risk, workload, and presence of mental disorders and neck and lower back pain. Dentists and more experienced PHC workers most frequently reported neck pain within the previous 12 months. Female workers most frequently reported lower back pain within the previous 12 months, and self-reported neck and lower back pain was associated with the level of frustration, ergonomic occupational risks, and the presence of mental disorder symptoms. These results show that the prevalence of neck and lower back pain coexists with working conditions that must be improved to ensure safe and healthy working environments. Additionally, the strategies the workers reported to manage pain require a multidisciplinary dialogue to reassess these strategies, mainly considering the risk of self-medication, one of the strategies reported by PHC workers. Likewise, local managers must devise and implement protective, preventive, and assistive measures within the PHC network.

## Figures and Tables

**Figure 1 healthcare-11-00365-f001:**
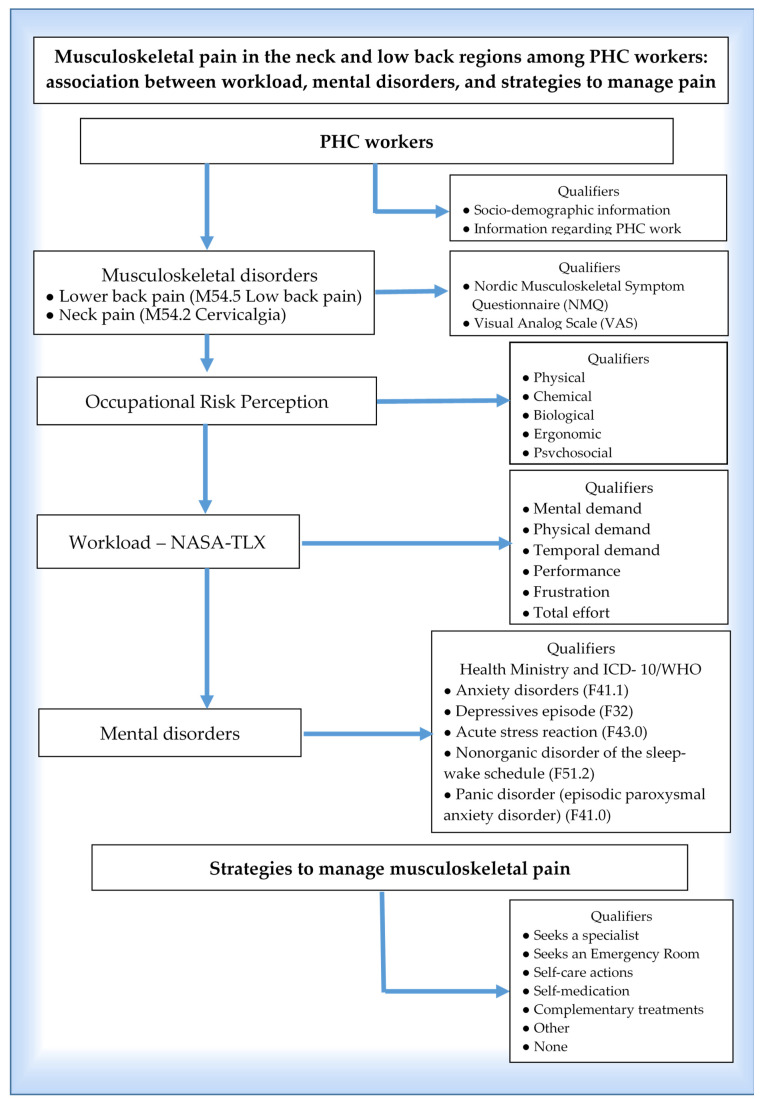
An outline of the key concepts integrated into musculoskeletal pain (M54.5 lower back pain; M54.2 Cervicalgia) and operational qualifiers adopted in this study.

**Figure 2 healthcare-11-00365-f002:**
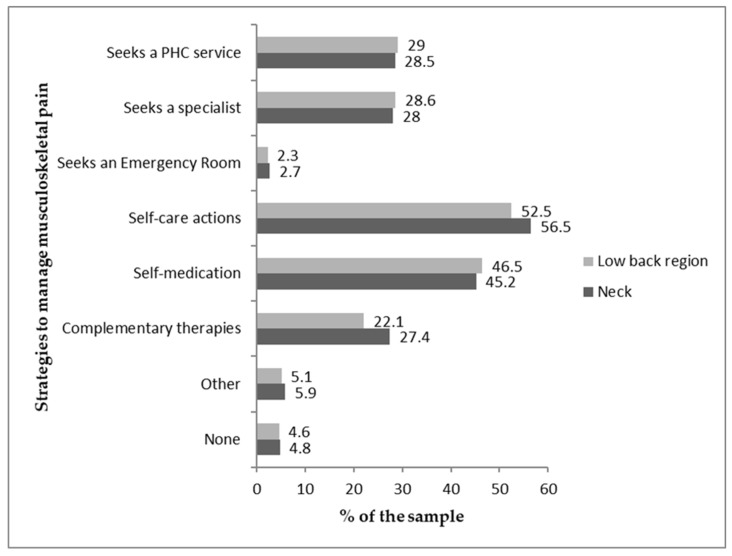
Sample distribution concerning strategies to manage musculoskeletal pain.

**Table 1 healthcare-11-00365-t001:** Association between variables and neck and lower back pain reported by PHC workers within the previous 12 months.

Variables	Total Sample n (%)	Neck Pain within the Previous 12 Months		*p*	Lower Back Pain within the Previous 12 Months		*p*
		Yes (n = 187)	No (n = 151)		Yes (n = 218)	No (n = 120)	
		n (%)	n (%)		n (%)	n (%)	
Age (years) *	41.4 ± 9.9	41.5 ± 9.8	41.2 ± 10.1	0.766 ^a^	41.8 ± 10.0	40.7 ± 9.9	0.331 ^a^
Gender				0.283 ^b^			0.012 ^b^
Male	45 (13.4)	21 (11.3)	24 (15.9)		21 (9.7)	24 (20.0)	
Female	292 (86.6)	165 (88.7)	127 (84.1)		196 (90.3)	96 (80.0)	
Race				0.393 ^b^			0.537 ^b^
Caucasian ^#^	258 (77.2)	143 (76.9)	115 (77.7)		170 (78.3)	88 (75.2)	
Afro-descendant	37 (11.1)	18 (9.7)	19 (12.8)		21 (9.7)	16 (13.7)	
Mixed	39 (11.7)	25 (13.4)	14 (9.5)		26 (12.0)	13 (11.1)	
Marital status				0.120 ^b^			0.462 ^b^
Single	99 (29.3)	46 (24.6)	53 (35.1)		62 (28.4)	37 (30.8)	
Married/Consensual union	198 (58.6)	120 (64.2)	78 (51.7)		125 (57.3)	73 (60.8)	
Separated/Divorced	36 (10.7)	18 (9.6)	18 (11.9)		27 (12.4)	9 (7.5)	
Widowed	5 (1.5)	3 (1.6)	2 (1.3)		4 (1.8)	1 (0.8)	
Educational level				0.371 ^b^			0.858 ^b^
Completed high school	129 (38.2)	70 (37.4)	59 (39.1)		85 (39.0)	44 (36.7)	
Some undergraduate studies/bachelor’s degree/technician	137 (40.5)	72 (38.5)	65 (43.0)		86 (39.4)	51 (42.5)	
specialization/Master’s/Ph.D.	72 (21.3)	45 (24.1)	27 (17.9)		47 (21.6)	25 (20.8)	
Number of children **	1 (0–2)	1 (1–2)	1 (0–2)	0.399 ^c^	1 (0–2)	1 (0–2)	0.115 ^c^
BMI (kg/m^2^) *	28.9 ± 5.9	29.1 ± 6.5	28.6 ± 5.2	0.528 ^a^	29.3 ± 6.5	28.1 ± 4.7	0.043 ^a^
Monthly income				0.521 ^b^			0.397 ^b^
Up to 2 times the m.w.	130 (39.2)	76 (41.3)	54 (36.5)		80 (37.2)	50 (42.7)	
2 to 4 times the m.w.	115 (34.6)	59 (32.1)	56 (37.8)		80 (37.2)	35 (29.9)	
>4 times the m.w.	87 (26.2)	49 (26.6)	38 (25.7)		55 (25.6)	32 (27.4)	
Location				0.663 ^b^			0.621 ^b^
Rio Grande	282 (83.4)	158 (84.5)	124 (82.1)		184 (84.4)	98 (81.7)	
São José do Norte	56 (16.6)	29 (15.5)	27 (17.9)		34 (15.6)	22 (18.3)	
Second job besides PHC	62 (18.4)	32 (17.1)	30 (20.0)	0.590 ^b^	38 (17.4)	24 (20.2)	0.590 ^b^
Profession				0.051 ^b^			0.864 ^b^
Nurse	50 (14.8)	27 (14.4)	23 (15.2)		32 (14.7)	18 (15.0)	
Physician	43 (12.7)	19 (10.2)	24 (15.9)		27 (12.4)	16 (13.3)	
Nursing technician/assistant	72 (21.3)	34 (18.2)	38 (25.2)		45 (20.6)	27 (22.5)	
Community health agent	139 (41.1)	83 (44.4)	56 (37.1)		89 (40.8)	50 (41.7)	
Dentist	13 (3.8)	10 (5.3)	3 (2.0)		11 (5.0)	2 (1.7)	
Oral health technician/assistant	15 (4.4)	8 (4.3)	7 (4.6)		10 (4.6)	5 (4.2)	
Other	6 (1.8)	6 (3.2)	0 (0.0)		4 (1.8)	2 (1.7)	
Years of experience in the profession **	11 (3–16)	11.5 (7–16.5)	10.3 (1.9–16)	0.125 ^c^	11.5 (7–16.5)	10.3 (1–15.4)	0.082 ^c^
Years working in PHC services **	8 (1–16)	9 (1.9–14.4)	6 (0.8–11.6)	0.007 ^c^	8.4 (1.8–13.1)	6 (0.7–11.9)	0.047 ^c^
Weekly workload	44.5 ± 13.2	39.7 ± 4.8	39.5 ± 5.8	0.718 ^a^	44.0 ± 13.4	45.3 ± 12.9	0.379 ^a^
Workday at PHC				0.114 ^b^			0.932 ^b^
Day shift	300 (89.3)	172 (92.5)	128 (85.3)		193 (88.9)	107 (89.9)	
Night shift	8 (2.4)	2 (1.1)	6 (4.0)		6 (2.8)	2 (1.7)	
Night/Day shift	23 (6.8)	9 (4.8)	14 (9.3)		15 (6.9)	8 (6.7)	
Other	5 (1.5)	3 (1.6)	2 (1.3)		3 (1.4)	2 (1.7)	
Workload *							
Mental demand	15.4 ± 5.2	16.1 ± 4.7	14.7 ± 5.7	0.015 ^a^	15.9 ± 5.0	14.6 ± 5.6	0.024 ^a^
Physical demand	12.5 ± 5.6	12.9 ± 5.3	11.9 ± 5.9	0.121 ^a^	12.4 ± 5.3	12.6 ± 6.1	0.718 ^a^
Temporal demand	14.1 ± 5.4	14.6 ± 5.0	13.5 ± 5.8	0.065 ^a^	14.3 ± 5.2	13.8 ± 5.7	0.387 ^a^
Performance	14.5 ± 5.4	14.5 ± 5.3	14.4 ± 5.5	0.859 ^a^	14.6 ± 5.3	14.3 ± 5.5	0.659 ^a^
Total effort level	15.3 ± 4.7	15.7 ± 4.3	14.7 ± 5.0	0.061 ^a^	15.5 ± 4.4	14.8 ± 5.1	0.137 ^a^
Frustration level	12.2 ± 6.0	13.6 ± 5.6	10.5 ± 6.2	<0.001 ^a^	12.8 ± 5.7	11.1 ± 6.5	0.014 ^a^
Occupational risks							
Physical	304 (90.7)	175 (94.6)	129 (86.0)	0.012 ^b^	204 (94.9)	100 (83.3)	0.001 ^b^
Chemical	269 (80.5)	159 (85.0)	110 (74.8)	0.028 ^b^	185 (85.6)	84 (71.2)	0.002 ^b^
Biological	317 (94.1)	179 (95.7)	138 (92.0)	0.228 ^b^	210 (96.8)	107 (89.2)	0.010 ^b^
Ergonomic	296 (87.8)	175 (93.6)	121 (80.7)	0.001 ^b^	203 (93.1)	93 (78.2)	<0.001 ^b^
Psychosocial	325 (96.2)	183 (97.9)	142 (94.0)	0.126 ^b^	214 (98.2)	111 (92.5)	0.015 ^b^
Psychiatric symptoms/disorders ***	241 (71.3)	157 (84.0)	84 (55.6)	<0.001 ^b^	175 (80.3)	66 (55.0)	<0.001 ^b^
Anxiety	175 (51.8)	117 (62.6)	58 (38.4)	<0.001 ^b^	129 (59.2)	46 (38.3)	<0.001 ^b^
Depressive episodes	97 (28.7)	73 (39.0)	24 (15.9)	<0.001 ^b^	78 (35.8)	19 (15.8)	<0.001 ^b^
Panic syndrome	33 (9.8)	24 (12.8)	9 (6.0)	0.053 ^b^	29 (13.3)	4 (3.3)	0.006 ^b^
Stress	182 (53.8)	126 (67.4)	56 (37.1)	<0.001 ^b^	137 (62.8)	45 (37.5)	<0.001 ^b^
Sleep–wake cycle disorder	78 (23.1)	62 (33.2)	16 (10.6)	<0.001 ^b^	62 (28.4)	16 (13.3)	<0.001 ^b^

^#^ One Asian individual was included. ^a^ Student’s t-test; ^b^ Pearson’s Chi-square; ^c^ Mann–Whitney Test; * Described by the mean ± SD; ** Described by the median (25–75 percentiles); *** Involves the presence of any of the following mental disorders: anxiety, depressive episodes, panic syndrome, stress, or sleep–wake cycle disorder.

**Table 2 healthcare-11-00365-t002:** Poisson Regression analysis to assess independent factors associated with neck and lower back pain within the previous 12 months.

Outcome	Factors	Prevalence Ratio (95%CI)	*p*
Neck pain within the previous 12 months	Profession		
	Nurse	1.00 (0.73–1.36)	0.984
	Physician	1.00	
	Nursing technician/assistant	0.94 (0.64–1.39)	0.760
	Community health agent	1.04 (0.79–1.37)	0.788
	Dentist	1.67 (1.11–2.51)	0.014
	Oral health technician/assistant	1.31 (0.83–2.07)	0.246
	Other	2.36 (1.51–3.69)	<0.001
	Years working in a PHC service	1.02 (1.00–1.03)	0.038
	Workload		
	Frustration level	1.03 (1.01–1.04)	0.005
	Occupational risks		
	Ergonomic	1.67 (1.06–2.62)	0.026
	Psychiatric symptoms/disorders	1.88 (1.39–2.54)	<0.001
Lower back pain within the previous 12 months	Gender		
	Male	1.00	
	Female	1.40 (1.01–1.94)	0.042
	BMI (kg/m^2^)	1.01 (1.00–1.02)	0.073
	Occupational risks		
	Chemical	1.25 (0.98–1.59)	0.070
	Ergonomic	1.62 (1.06–2.48)	0.025
	Psychiatric symptoms/disorders	1.45 (1.16–1.82)	0.001

95%CI = 95% confidence interval.

## Data Availability

Data regarding this study can be provided upon request to the corresponding author. Data are not publicly available due to ethical issues.

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
