# Peer review of "Musculoskeletal Pain in the Neck and Lower Back Regions among PHC Workers: Association between Workload, Mental Disorders, and Strategies to Manage Pain"

_healthcare, 2023, doi:10.3390/healthcare11030365_

Round 1
Reviewer 1 Report
General comments:
The paper should be checked by a native speaker.
There is no consistency in your presentation. For example workers, healthcare workers, participants.....
The tense of the presentation is not similar in different sections and sentences.
I cannot see the novelty in your work unless you determine the novelty of your work clearly.
Abstract: Many details are missing in the method section.
The conclusion is not in line with your results.
Introduction:
It is very messy and difficult to follow. Most of the papers are from Brazil and Portugal and there are some biases. Do you mean there is no other valuable and related research from other countries??
Your introduction is not following the standard format and it is such a literature review. You need to rewrite it fully.
It is very confusing because you are talking about different terms. Musculoskeletal injuries, musculoskeletal disorders, musculoskeletal system, musculoskeletal low back pain, musculoskeletal diseases, musculoskeletal symptoms, etc... .Please ask an expert to rewrite it. Those terms have different meanings. and it should be corrected in the whole paper.
"Musculoskeletal disorders represent a substantial burden of diseases, being the 60 leading cause of injuries worldwide, affecting populations of different ages, both youths, 61 and adults" is a copy from the abstract.
Methods:
Section 2.2., Measures is too long and very difficult to understand and follow.
Result: Please replace sex with gender.
The result should be separated from the discussion.
Once you are talking about issues in your result section and do a comparison with previous studies. For example, maybe the MSD could be related to the lack of exercise. So, you need to add those topics as well. Please have a look at Office exercise training to reduce and prevent the occurrence of musculoskeletal disorders among office workers: a hypothesis. The Malaysian journal of medical sciences: MJMS, 23(4), 54.
Overall, the work seems good, but the type of presentation needs to some changes. The writing is too long and has too many tables.
--
Author Response
Dear Reviewer,
We greatly appreciate your comments. We complied with the recommendations and made improvements to the manuscript. We are writing to provide point-by-point responses to your comments and suggestions.
We are grateful for the careful work of revising the manuscript.

Reviewer 2 Report
Dear Corresponding Author,
Thank you for submitting your paper. It is a very relvant and practical issue.
I read your paper and I point out some suggestions:
1) Abstract: you introduce directly PHC as acronym without using the full description, it is not easy for the reader; lines 27 and 28 delete M54.2 and.5 because there is no explanatio in the abstract; line 34 delete ">8 points", it is unuseful.
2) It's my fault but I did not know the international classification code of the pain M54. I found it on the web, maybe it is better to write a clarification in line 54.
3) The first part of the introduction is very interesting but it is hard to read because there are too many acronyms. In the rest of the paper you don't use so much "DALYs", "YLLs", "YLD" therefore I suggest to use the full name. It is easier for the reader. Moreover, in line 70 you write "GDP" but it is the first time that appears on the paper, please use the full name.
4) Adjust the citation format in line 87
5) I suggest to add some lines for describing a list of typical PHC workers. It could be helpul for a better understanding of the target involved.
6) In line 163 you declare 1 physical theapist and 1 physical educator. In the abstract, in line 36 you use the plural, why? More over the results in line 36 could not be possibile because you have only 1 subjects for each of those two professions. As well as in lines 376-377; 442-446. If your sample has just one physical therapist it is not consistent to adress these results. I suggest to cancel this confunding information.
7) The lines 164-167 are not very clear. What do you mean when you write that the last day you add new participants? How many? It is not clear for me.
I appreciated your paper, globally. The limits sound good and the discussion is oriented on a practical direction. A better evaluation of the physical activity levels should have been very useful and interesting but actually the study doesn't have this information. The main problem that I suggest to correct is the discussion about the physical therapists and educators. You have a good sample sizie, well distributed with several professions and then you have just 6 "other" professions. Please delete these 6 and don't use them for the analysis because the numerosity is too low and unuseful. Delete the information that you wrote on physical therapists because you have only 1 physical therapist and it is not correct what you wrote in the abstract and in the discussion about that professional.
Author Response

(The authors gave the same response as above.)

Round 2
Reviewer 1 Report
I am happy about this revised version.
Author Response
Dear Reviewer,
We appreciate your expert and thoughtful feedback to improve our manuscript. We are academically pleased to be able to meet your recommendations. Stay well.
Corresponding Author and (Co-Author)
Reviewer 2 Report
Dear Corresponding Author,
thanks for your comments. I read again the paper and it sounds improved. Congratulation. Nevertheless, in the following lines you can find some comments.
1) PHC in the abstract is abbreviated without description in line 26. Please use the full name for the first time and then use the acronym.
2) It's not clear for me why you totally changed the introduction. It was better the previous one. The new one is shorter and smart but it is a little bit confused. In lines 83-85 you write something that is connected with the aim of the study but then you continue writing other information. I suggested to read it carefully.
I think I have to inform the Editor that you totally changed the introduction. It is not a common practice. I am not used to write comments to the Editor but, you know, you can understand that this situation requires a comment.
3) I am not sure that the way you cite the literature in line 138 is correct, it is better to ask to the editor.
4) The sentence in lines 138-139 could be deleted.
Author Response
Dear Reviewer,
We appreciate your expert and thoughtful feedback to improve our manuscript. We are writing to provide point-by-point responses to your comments and suggestions. We hope we have responded to the recommendations. Stay well.
Corresponding Author and (Co-Author)
